# Engineering Strategies to Enhance TCR-Based Adoptive T Cell Therapy

**DOI:** 10.3390/cells9061485

**Published:** 2020-06-18

**Authors:** Jan A. Rath, Caroline Arber

**Affiliations:** Department of oncology UNIL CHUV, Ludwig Institute for Cancer Research Lausanne, Lausanne University Hospital and University of Lausanne, 1015 Lausanne, Switzerland; jan.rath@unil.ch

**Keywords:** adoptive T cell therapy, transgenic TCR, engineered T cells, avidity, chimeric receptors, chimeric antigen receptor, cancer immunotherapy, CRISPR, gene editing, tumor microenvironment

## Abstract

T cell receptor (TCR)-based adoptive T cell therapies (ACT) hold great promise for the treatment of cancer, as TCRs can cover a broad range of target antigens. Here we summarize basic, translational and clinical results that provide insight into the challenges and opportunities of TCR-based ACT. We review the characteristics of target antigens and conventional αβ-TCRs, and provide a summary of published clinical trials with TCR-transgenic T cell therapies. We discuss how synthetic biology and innovative engineering strategies are poised to provide solutions for overcoming current limitations, that include functional avidity, MHC restriction, and most importantly, the tumor microenvironment. We also highlight the impact of precision genome editing on the next iteration of TCR-transgenic T cell therapies, and the discovery of novel immune engineering targets. We are convinced that some of these innovations will enable the field to move TCR gene therapy to the next level.

## 1. Introduction

Adoptive T cell therapy (ACT) with T cells expressing native or transgenic αβ-T cell receptors (TCRs) is a promising treatment for cancer, as TCRs cover a wide range of potential target antigens [1]. Native TCR specificities have successfully been exploited for ACT with tumor infiltrating lymphocytes (TILs) for melanoma [2] and other tumors [1], or with virus-specific T cells (VSTs) for viral-associated malignancies [3]. Transgenic TCR-based ACT allows the genetic redirection of T cell specificity in a highly specific and reproducible manner, and has produced promising results in melanoma and several solid tumors [4,5], multiple myeloma (MM) [6,7], viral-associated malignancies [8] and acute myeloid leukemia (AML) [9]. Even though clinical results are encouraging for both approaches, several major limitations have been identified and are being addressed by various engineering strategies. Those challenges include: target antigen selection, TCR selection, human leukocyte antigen (HLA) restriction, antigen escape, T cell homing to the tumor, T cell infiltration into the tumor, T cell persistence, and local immunosuppression in the tumor microenvironment (TME) [1,10,11]. In this review, we will highlight recent progress in the field and focus on innovative engineering strategies with the potential to deliver effective and long-lasting tumor control to a broad range of cancer patients.

We will not discuss chimeric antigen receptor (CAR)-based T cell therapies and recent developments of these unless specific co-engineering strategies have important implications for the TCR field. Approaches that involve the combination of TCR-based ACT with drugs, peptides or dendritic cell vaccines, and the historical context of ACT, have been extensively reviewed elsewhere [12,13,14,15,16,17,18,19,20,21].

## 2. Targeting Tumors with TCR-Based ACT

### 2.1. Antigen Recognition by TCRs

The TCR is a heterodimer consisting of the TCRα- and β-chains that recognize the target antigen with their variable regions in the context of a specific peptide-major histocompatibility complex (pMHC). The constant regions associate with the CD3 chains (γ, δ, ε, ζ) to form the functional TCR complex (Figure 1). TCRs exhibit a broad range of affinities for the targeted pMHC on the surface of target cells. Upon engagement of the TCRαβ with the cognate pMHC, the associated CD3 complex initiates the signaling cascade needed for T cell activation (signal 1). Full deployment of T cell function, expansion, memory formation and long-term persistence depend on the presence of co-stimulation (signal 2) and cytokine signals (signal 3) [22,23]. These additional signals are available to the T cells when they encounter professional antigen-presenting cells, but are reduced or absent in the TME [24]. The strength of the TCR:pMHC interaction is a crucial parameter in TCR-based T cell therapies, as it governs the functional response of T cells (discussed in Section 2.3). Conventional TCR-based ACTs are MHC-restricted, which limits the applicability of the approach. Mechanistic details of TCR structure, signaling, and binding to pMHC have previously been discussed [25,26].

### 2.2. Target Antigen Selection for TCR Gene Therapy

A major challenge in the development of αβ-TCR-based ACT is the identification of suitable target antigens that are presented as peptides in the context of specific HLA alleles [27]. The features of an ideal tumor-associated antigen (TAA) for safe and effective targeting include: (1) selective expression on tumor cells, (2) absence or low expression on healthy cells, (3) sufficient immunogenicity to mediate T cell responses, and (4) dependency of tumor survival on the antigen (e.g., oncogene). Unfortunately, only very few known target antigens match these criteria, and the frequencies of the restricting HLA alleles further narrow down the choice. The spectrum of αβ-TCR amenable targets are traditionally classified into five groups with increasing specificity for tumor cells: (1) ubiquitous antigens (e.g., Her2/neu), (2) overexpressed self-antigens (e.g., Wilms’ Tumor antigen 1, WT1; survivin), (3) differentiation antigens (e.g., tyrosinase), (4) cancer testis antigens (CTAs, e.g., melanoma-associated antigen, MAGE; New York esophageal squamous cell carcinoma-1, NY-ESO-1), and (5) tumor-specific antigens (TSA, e.g., viral-derived oncoproteins, cancer neoantigens) [27,28,29,30]. Two additional types of target antigens with interesting features are emerging in the field: human endogenous retroviruses (hERVs) and MHC-independent antigens. We will highlight recent progress in neoantigen identification and targeting, and briefly discuss the two types of emerging alternative targets.

While CTAs have already been intensively investigated clinically, especially the NY-ESO-1 antigen (see also Section 2.4), cancer neoantigens have now advanced to become a prime target for the development of TCR-based therapies. Their importance has recently been reviewed and discussed in great detail [31]. Neoantigens derived from cancer-specific genetic alterations mostly yield patient-specific private antigens. However, some can be derived from common chromosomal translocations or oncogenic driver mutations, and can be shared by several tumor types in the context of frequent HLA alleles (also called “public neoantigens”). Tumor mutational burden (TMB) varies among tumor histologies [32], and high TMB is likely linked to high neoantigen load. For example, strong polyclonal T cell responses can be unleashed upon immune checkpoint inhibitor treatment in some of these high TMB patients [33]. On the other hand, low TMB does not preclude the presence of immunogenic neoantigens, as demonstrated, for example, in pediatric acute lymphoblastic leukemia [34]. As mentioned above, target antigen selection for transgenic TCR-based ACT focuses on specific single epitopes. Hence, the challenge lies in selecting a relevant antigen among the many that are present in a given tumor, and identifying a suitable TCR that recognizes the cognate pMHC with optimal affinity. To tackle this challenge, high-throughput neoantigen discovery pipelines have been developed [35] that include next-generation sequencing and single-cell RNA sequencing for TCR identification [36], novel epitope prediction methods [37,38,39,40], mass-spectrometry [41,42] and T cell-based validation assays [43,44,45]. Such pipelines help significantly in advancing the discovery process for the efficient detection of immunogenic neoantigens and matching TCR sequences [46,47]. Once a suitable TCR is identified, neoantigen targeted TCR-based ACTs may be applied as a highly personalized therapy [29,31], or more broadly proposed when targeting public neoantigens [48,49,50,51]. Targeting epitopes derived from true oncogenic drivers is of particular interest, since the likelihood of therapy resistance due to antigen escape upon ACT is predicted to be low. Clinical trials investigating TCRs targeting neo-epitopes, such as G12V (NCT03190941) or G12D (NCT03745326) mutant Ras GTPase (RAS) in the context of HLA-A*11:01, or undisclosed targets (NCT03412877, NCT04102436, NCT03970382), are currently recruiting.

Two additional types of emerging target antigens include hERVs- and MHC-independent antigens. The features of hERVs are quite similar to CTAs, as they are rarely expressed in normal adult tissues but are upregulated upon epigenetic dysregulation [30]. A potential role for them in eliciting tumor-specific T cell responses has been discussed in mouse and human tumor models, but we still need to learn more about their expression across different tumor histologies and normal tissues. A clinical trial investigating an HLA-A*11:01-restricted hERV epitope for treatment of clear cell renal cell carcinoma is recruiting patients (NCT03354390). MHC-independent antigens are not peptide- but lipid- or metabolite-derived, and are presented in the context of the non-polymorphic CD1 or MHC-related molecule 1 (MR1) [52]. These antigens can be targeted with unconventional αβ-TCRs that are not MHC-restricted [53,54]. The anti-tumor activity of CD1d-restricted lipid targeted αβ-TCR invariant natural killer T cells (iNKT) cells has already been established in preclinical model systems, and is also being investigated clinically, for example for hepatocellular carcinoma (NCT04011033). iNKT cells can also serve as a platform to express CARs, such as targeting CD19 for B cell lymphoma [55]. Clinical translation of this approach is ongoing (NCT03774654). The anti-tumor potential of MR1-restricted αβ-TCR T cells is just emerging, and yet to be characterized.

### 2.3. TCR Affinity

TCR affinity describes the physical binding strength of the TCR towards the cognate pMHC in cell-independent conditions [56]. The affinity of a TCR is inversely proportional to the dissociation equilibrium constant K_D_, that is commonly determined by surface plasmon resonance (SPR), and describes the ratio of TCR:pMHC complex association (K_on_) and dissociation (K_off_). Naturally occurring TCRs that have undergone thymic selection are usually of low affinity against self TAAs, in contrast to TCRs targeting viral or neoantigens [57,58]. Low affinity TCRs targeting TAAs have produced limited anti-tumor function and proliferation in CD8+ T cells, and lack capability to redirect CD4+ T cells to MHC class I-restricted epitopes [59,60,61,62], which requires higher TCR affinities than most naturally occurring TCRs [62,63,64,65].

TCR affinity can be modulated by engineering the complementarity determining regions (CDRs) involved in the binding of the cognate pMHC. High-affinity TCRs were generated with yeast or phage display [66,67,68,69], computational modeling [70,71], or a combination of methods [72], and produced TCR transgenic T cells with potent anti-tumor activity [5,73]. However, supraphysiologic TCR affinities have been shown to impair T cell function, suggesting that optimal affinity thresholds should be considered when developing TCR-engineered T cells [74,75,76]. In addition, modification of pMHC binding properties of a given TCR can cause unpredicted cross-reactivities against self-antigens, and can lead to severe, sometimes lethal, toxicities in patients [77,78,79]. Therefore, novel, more stringent preclinical screening algorithms to identify potential on- and off-target toxicities of affinity-engineered TCRs have been developed [80]. Further development and refinement of combined computational and experimental screening strategies will be required for safe clinical translation of affinity-enhanced TCRs.

### 2.4. TCR Gene Therapy Clinical Results

Clinical ACT trials with TCR transgenic T cells have so far evaluated the safety and anti-tumor function of T cells directed against differentiation antigens in melanoma (melanoma antigen recognized by T-cells 1, MART-1; tyrosinase/gp100) [81,82,83] and colorectal cancer (carcinoembryonic antigen, CEA) [84], CTAs (NY-ESO-1, MAGE) in a variety of solid tumors and multiple myeloma [4,5,6,7,77,79,85,86,87,88,89], the overexpressed self-antigen WT-1 in myeloid malignancies [9,90], and Human Papilloma Virus 16 (HPV-16)-associated antigen E6 in epithelial cancers [8]. All but one of these TCRs were HLA class I-restricted, and the only class II-restricted TCR was directed against MAGE-A3 [86]. An overview of published clinical trials is given in Table 1.

TCR sequences used in these trials originated from healthy human donors, TILs of cancer patients or vaccinated HLA transgenic mice. Some of the TCRs were ex vivo engineered, high-affinity variants with improved anti-tumor function [4,5,6,73,77,78,79,84,87,88,91,92]. Of these affinity-engineered TCRs, unfortunately only the HLA-A*02:01-restricted NY-ESO-1 1G4-a95:LY or c259 TCR proved safe [4,5], while the other three TCRs produced significant, and in some cases lethal, toxicities [9,77,79,83,84]. Specifically, dose limiting on-target off-tumor toxicity with inflammatory colitis and diarrhea was observed upon infusion of a CEA-specific TCR T cells in two of three patients [84]. Severe, and in two cases lethal, neurotoxicity was seen upon treatment with a MAGE-A3/12-specific HLA-A*02:01-restricted TCR, and attributed to the previously unrecognized expression of MAGE-A12 in a subset of neurons in the human brain [79]. Lethal off-target cardiotoxicity occurred in two patients treated with a MAGE-A3 HLA-A*01:01-restricted TCR, due to cross-reactivity and recognition of titin in beating cardiomyocytes [77]. These severe toxicities led to the development of novel preclinical screening pipelines for affinity-enhanced TCRs, in order to better identify potential on- and off-target reactivities before selecting a candidate TCR for clinical translation [80]. Incorporation of genetic safety switches in TCR-engineered T cells will be discussed in Section 3.

All trials conducted in patients with melanoma or solid tumors included heavily pretreated patients who were refractory to available standard treatments, a major challenge for the infused T cells to efficiently deploy their full therapeutic potential. The administration of lymphodepleting chemotherapy and high-dose IL-2 after ACT proved beneficial to achieving responses. In contrast, in the absence of lymphodepletion, stable disease was the best response obtained [85,90]. TCRs targeting differentiation antigens (MART-1, tyrosinase/gp100, CEA) were mostly limited by on-target, off-tumor toxicities to healthy tissues that share antigen expression with the tumor. Therefore, TCRs that target antigens more restricted to tumor cells may be better tolerated by the patients. CTA NY-ESO-1-specific TCR T cell ACT was very well tolerated and produced promising overall response rates (ORR) in refractory synovial sarcoma patients across several studies [4,5,6,87,88]. For example, an overall response rate of 61% (11/18) was reported in patients with synovial sarcoma, and 55% (11/20) in patients with melanoma, but most responses lasted less than 1 year [4,5]. In a subsequent independent trial, an ORR of 50% was achieved (6/12 patients), with a median duration of response of 30.9 weeks in patients with synovial sarcoma [87]. In this trial, that omitted high dose IL-2 administration, NY-ESO-1 TCR T cells expanded significantly and persisted long-term in the peripheral blood of responding patients. T cells with an early differentiated phenotype were enriched in the persisting population, and tissue infiltration of TCR+ T cells was demonstrated in some cases. However, T cell persistence in peripheral blood did not necessarily protect from relapse. One patient with a late relapse showed no evidence of T cell infiltration in the tumor, indicating that absence of T cell migration/penetration into the site of tumor recurrence was the mechanism for relapse, while the tumor cells still expressed the NY-ESO-1 antigen. The fact that 120 patients were screened for this trial and only 12 patients received the treatment highlights the challenge of HLA restriction to efficiently deliver HLA-restricted, TCR-transgenic T cell therapies to patients. Recently, a trial with multiplexed genome editing of T cells and NY-ESO-1 TCR transduction demonstrated the safety and feasibility of the approach in three patients (discussed in more detail in Section 5.2) [89].

Three trials focused on patients with hematologic malignancies. In the first trial, autologous affinity-engineered NY-ESO-1 TCR-specific T cells were given safely to patients with advanced MM two days after high-dose melphalan and autologous hematopoietic stem cell transplantation (ASCT), followed by lenalidomide maintenance [6]. The outcomes compared favorably to historic controls. In the study, 70% of the 20 patients treated achieved near complete response (nCR) or complete response (CR) [6], while in cohorts of patients with less aggressive disease, response rates were typically <40% post-transplant [93]. However, the exact impact of the infused TCR+ T cells still needs to be investigated, as in this study it was difficult to definitively discern TCR+ T cell effect from the high-dose melphalan, ASCT and lenalidomide maintenance effect. The second trial evaluated an HLA-A*24:02-restricted WT-1 TCR in patients with AML or myelodysplastic syndrome (MDS) [90]. T cell infusions were safe, but in the absence of lymphodepletion, the responses were disappointing. Lastly, another trial assessed safety and efficacy of allogeneic WT-1 TCR-specific, Epstein Barr Virus-specific CD8+ T cells (EBVSTs) derived from the allogeneic stem cell donor [9]. TCR+ EBVSTs were safely infused to high-risk AML patients with no evidence of disease after allogeneic hematopoietic stem cell transplantation (HSCT), followed by low dose IL-2 and a second infusion in 7 of 12 patients. Toxicities were manageable, and at a median follow up of 44 months, all 12 patients remained relapse-free. In a matched control cohort of 88 patients treated at the same institution, relapse-free survival was 54%. These results indicate that transgenic TCR-based T cell therapy is highly promising for high-risk AML patients, when given in the post-transplant setting and undetectable disease state.

These clinical results indicate that several factors are crucial for the success of TCR gene therapy: the selection of the targeted antigen, the specific features of the TCR used, the biology of the T cell in which the TCR is expressed, the disease status of the patient, the administration of lymphodepleting chemotherapy and in vivo cytokine support of the infused cells, the in vivo persistence, the homing to and infiltration of the tumor, the impact of the TME on T cells that successfully reached the TME, and the overall safety of the approach. Engineering strategies to address these challenges will be discussed next.

## 3. Engineering Safety

ACT, including TCR-based approaches, can produce severe and life-threatening toxicities (as discussed in Section 2.4 and summarized in Table 1) [102]. Thus, engineering safety into therapeutic T cells, with drug controllable switches that allow for the rapid elimination of T cells in vivo, have been developed (reviewed in [21]). Three main strategies have been investigated clinically, (1) the herpes simplex virus thymidine kinase (HSV-TK) suicide gene, (2) the inducible caspase 9 (iCasp9 or iC9) suicide gene, and (3) cellular elimination tags (CET) (Figure 2a–c).

T cells genetically engineered to express the HSV-TK can be specifically ablated by administration of the pro-drug ganciclovir (Figure 2a). HSV-TK converts ganciclovir into metabolites that block DNA synthesis, leading to apoptosis of dividing cells. HSV-TK-engineered donor lymphocyte infusions (DLIs) were administered to patients after allogeneic HSCT in several clinical trials, to enhance graft-versus-leukemia effects while limiting graft-versus-host disease (GVHD) [103,104,105,106]. Drawbacks are the slow mechanism of action (dependence on cell division), immunogenicity of the transgene and toxicity of the prodrug ganciclovir. Nevertheless, HSV-TK is the most clinically investigated genetic safety system to date.

A safe, reliable and more rapid genetic approach for the drug-inducible removal of transgenic T cells is the iC9 safety system (Figure 2b). iC9 is a human fusion protein consisting of a truncated caspase 9 linked to a modified FK506- binding protein (FKBP-F36V) [107]. Dimerization of the fusion protein by the inert small molecule AP1903/Rimiducid rapidly initiates the apoptosis of transgenic cells through the intrinsic (mitochondrial) caspase 3 apoptotic pathway. Activation of iC9 can control toxicities upon ACT, such as the development of GVHD after allogeneic HSCT and DLI [108,109,110,111].

Cellular elimination tags (CET) consisting of a truncated marker protein, such as tEGFR, tCD20 or the CD34-CD20 fusion RQR8, are also used clinically, as they can be targeted by clinically approved antibodies (Figure 2c). Target cell depletion is achieved through antibody-dependent cellular cytotoxicity [112,113,114]. Several clinical trials with CETs are ongoing, but formal demonstration of their clinical impact in controlling toxicities remains to be reported.

Other elimination strategies aiming at drug controllable proteasomal degradation of the transgene have been developed preclinically (Figure 2d). For instance, a CAR can be degraded upon administration of a hepatitis C virus (HCV) protease inhibitor (e.g., asunaprevir) if the CAR is linked to the HCV NS3 protease target site, followed by the NS3 protease and a degradation moiety (degron) [115]. In the absence of the drug, the degradation moiety is cleaved and the CAR stably expressed. Such systems could also be evaluated with transgenic TCRs.

## 4. T Cell Engineering Strategies to Enhance Anti-Tumor Activity of TCR-Transgenic T Cells

### 4.1. Enhancing Functional Avidity

A critical determinant for TCR-engineered T cell function is the initiation of a functional response towards a defined concentration of cognate pMHC (e.g., target killing, proliferation, cytokine production). Common terms used to describe this property include antigen sensitivity and functional avidity [116,117,118], defined by physical parameters, such as TCR affinity and avidity (multimeric binding strength on a cell), and several biological parameters that include: TCR density [119,120,121] and clustering [122,123,124,125], T cell activation [126,127,128] and differentiation status [129], glycosylation [130,131,132], expression of co-stimulatory or inhibitory receptors [133,134,135,136,137,138], adhesion molecules [139,140,141] and availability of TCR complex signaling components (e.g., CD3) [142] and co-receptors (e.g., CD8αβ) [143]. For class I restricted TCRs, TCR affinity is the major factor that determines whether or not the TCR depends on the presence of the CD8αβ co-receptor for target antigen recognition and the deployment of effector function [144]. The CD8αβ co-receptor has several critical roles in the TCR–pMHC interaction which can enhance antigen sensitivity over several log fold [144,145]. The cooperative binding of CD8 with the TCR to pMHC, for example, results in the stabilization of the TCR–pMHC complex by lowering the dissociation rate [143,146,147]. CD8αβ delivers key signaling components, such as LCK, to the cytoplasmic side of the TCR/CD3 complex [148,149], and enables efficient signal transduction through partitioning the TCR in the optimal membrane compartment [149,150,151].

Taken together, these observations have implications for the engineering of TCR-transgenic T cells. First, mispairing of introduced and endogenous TCRα and β chains must be prevented (reviewed in [152]). In brief, the explored strategies mostly focus on the constant regions and include replacement of human with murine constant regions [153], minimal murinization [154], codon optimization [155,156], addition of disulfide bonds [157,158], incorporation of hydrophobic mutations [159], removal of N-glycosylation sites [160], TCR domain swapping [161], expression of the transgenic TCR as TCR–CD3ζ fusion protein [162], use of a single chain TCR format [163,164], or silencing or genetic knockout of endogenous TCR chains (see also Section 5.1) [165,166,167,168]. More recently, TCR framework engineering in the variable regions that does not contribute to antigen recognition was shown to safely enhance the functional avidity of TCR-transgenic T cells [169]. Second, upregulation of specific adhesion molecules has the potential to enhance T cell-target cell interactions. For example, forced expression of an engineered shedding-resistant version of L-selectin (CD62L) has been shown to promote persistence and anti-tumor function of mouse TCR T cells in a melanoma model [170]. Third, limited availability of and competition for endogenous CD3 signaling chains can reduce transgenic TCR function. Co-transferring the four CD3 chains together with a transgenic TCR was shown to improve functional avidity of TCR-engineered T cells (Figure 3a) [142]. Fourth, several groups have demonstrated that CD4+ T cells redirected with a class I-restricted TCR and transgenic CD8αβ (TCR8) are hybrid TCR8+ CD4+ T cells, that are cytotoxic, and at the same time preserve their CD4+ lineage features (Figure 3a) [171,172,173,174,175,176]. We recently showed by single-cell RNA sequencing and experimental validation that TCR8+CD4+ T cells exhibited superior anti-tumor function in vitro and in vivo compared to TCR+ or TCR8+CD8+ T cells [177]. Transcriptional programs were related to enhanced T cell function, including proliferation, oxidative phosphorylation, cytotoxicity and co-stimulation, while differentiation and exhaustion programs remained low.

### 4.2. Engineering MHC-Independent Antigen Specificity through TCR

A major challenge in conventional αβ-TCR-based T cell therapies is their MHC restriction, which limits their applicability to a small pool of patients with the matching HLA allele. In fact, most TCRs investigated in the completed clinical trials are restricted to HLA-A*02:01 (Table 1). However, even when targeting peptides presented in the context of the highly frequent HLA-A*02:01 allele, only 10% of patients with synovial sarcoma screened actually received the treatment (12/120 patients) [87].

However, a subset of naturally occurring αβ-TCRs are MHC-independent, and therefore potentially more broadly applicable, including for example lipid-restricted (CD1) T cells or monomorphic MHC class I-related protein (MR1)-restricted T cells (reviewed in [53,178]). These HLA-independent TCRs recognize ligands related to tumor or pathogen-derived lipids, metabolites, phospho-antigens or stress-ligands, and are emerging as attractive alternatives for TCR-based therapies. For instance, CD1d-restricted iNKT cells efficiently recognize and kill universal lipid antigens presented on tumor cells [53]. MR1-restricted T cells mostly reside in mucosal tissues and specifically recognize components of pathogens. More recently, tumor-specific activity of an MR1-restricted TCR has been reported with activity against a wide variety of tumors, but not normal tissues or pathogen-derived antigens [52]. The exact target antigen is yet to be identified. 

Another strategy to mitigate MHC restriction while taking advantage of physiologic αβ-TCR signaling (unlike CAR T cells) is to redirect the αβ-TCR to cell surface antigens by introduction of antibody-derived recognition domains. For instance, the T cell antigen coupler (TAC) consists of a modified CD4 co-receptor, where the extracellular domain is replaced by two single chain variable fragments (scFv), that bind to a tumor antigen and CD3ε of the αβ-TCR complex respectively [129] (Figure 3b). A more simplified version is the TCR fusion complex (TRuC), which consists of a scFv fused directly to one of the CD3 chains (e.g., CD3ε) and forms a functional TCR complex (Figure 3b) [179]. In both approaches, the scFvs initiated efficient TCR signaling in the absence of co-stimulation, and independently of TCR specificity. Compared to selected second generation CAR T cells, TAC and TRuC expressing T cells exhibited superior anti-tumor function with a better safety profile due to lower cytokine production. Interestingly, TRuCs preserved the MHC-restricted antigen response, which opens the possibility of engineering TCR T cells with dual specificity for MHC-dependent and -independent antigens, and could be a strategy for counterbalancing antigen escape.

These intriguing findings and strategies point towards the potential of moving TCR-based ACTs beyond HLA restriction, opening the application to a much larger patient population with applicability to multiple cancer types.

### 4.3. Engineering Strategies to Target the Tumor Microenvironment

The TME represents a heterogenous mix of tumor-associated cells and extracellular components, which provide the tumor cells with a niche for enhanced survival and protection against antigen-specific T cells [180]. The limited success of ACT, particularly in solid tumors, is likely linked to the fact that infused T cells need to overcome multiple layers of challenges when encountering the TME [181]. First, the tumor stroma, which consists primarily of tumor-associated cells, represents a physical barrier which blocks efficient T cell infiltration [182]. Second, lack of T cell-specific homing factors in the TME leads to an undirected migration and subsequent loss of target cell detection [183]. Third, the TME is devoid of expression of co-stimulatory ligands and immune stimulatory cytokines, which are required for full T cell effector function and persistence. Lastly, T cell inhibitory ligands (immune checkpoints) are upregulated and bind to co-inhibitory receptors expressed on T cells. Immunosuppressive cells (e.g., regulatory T cells, tumor-associated macrophages, myeloid derived suppressor cells) infiltrate the TME, and further support effector T cell inhibition [180]. In the field of CAR T cell therapies, it has long been recognized that the incorporation of co-stimulatory endodomains is critical for the long-term persistence of adoptively transferred engineered T cells and lasting tumor control [184,185,186,187]. In addition, therapeutic strategies that help counterbalance the negative impact of the TME, such as immune checkpoint inhibitors, have become a cornerstone in oncology, and combinatorial strategies are actively investigated [16]. Over the past several years, various co-engineering strategies have been developed with the goal of enhancing the tumor infiltration and immunological properties of CAR T cells (recently reviewed in [20,21]). In the TCR T cell therapy field, however, improved tumor infiltration and simultaneous delivery of signals 2 (co-stimulation) and/or 3 (cytokine signal) to engineered T cells have been less extensively investigated. Strategies which enhance TCR transgenic T cell function through targeting the TME will be discussed below, and are summarized in Table 2.

#### 4.3.1. Engineering T Cell Homing and Tumor Infiltration

Limited tumor infiltration by adoptively transferred TCR T cells has been described as a mechanism of malignant relapse [87], suggesting that T cells lack specific homing and penetration capabilities to the tumor site. Therefore, enhancing the T cells’ capacity to home to and infiltrate tumors by genetic engineering has been investigated. For instance, elevated levels of CXCL8/IL-8 in the melanoma microenvironment [188] can be harnessed by engineering T cells to overexpress the matching chemokine receptor CXCR2 (Figure 3c). MAGE-A3 TCR transgenic T cells overexpressing CXCR2 exhibited superior tumor infiltration capacity in a xenograft mouse model of human melanoma [189].

Similarly, the co-expression of CCR2 and CCR4 in CAR T cells has been shown to improve homing to the tumor site and anti-tumor function in mouse models of malignant pleural mesotheliomas, Hodgkin’s lymphoma and neuroblastoma, respectively [190,191,192].

An additional challenge in solid tumors is the physical shielding of tumor cells by the surrounding tumor stroma, a heterogeneous mix of cells such as cancer-associated fibroblasts (CAFs) and extracellular matrix (ECM) components, including heparan sulfate proteoglycans (HSPGs). Extravasation of T cells to the tumor tissue can be enhanced by overexpressing ECM-degrading enzymes (Figure 3c). For instance, GD2-targeted CAR T cells engineered with heparanase (HPSE), an enzyme which degrades HSPGs of the ECM, have been shown to improve infiltration and anti-tumor function in a xenograft mouse model of human neuroblastoma [193]. Another promising strategy is the targeting of fibroblast activation protein-α (FAP) expressed on CAFs, to eliminate the tumor stroma and enabling infiltration into the tumor site (Figure 3c). Mouse or human T cells engineered with a CAR targeting FAP improved tumor control and T cell infiltration in vivo in various mouse tumor models [194,195,196].

#### 4.3.2. Delivery of Co-Stimulation

To address the problem of limited local or systemic persistence of adoptively transferred TCR+ T cells, T cells can be engineered to receive co-stimulation in various ways. For example, transgenic TCR chains can be engineered to incorporate the transmembrane and endodomain of the co-stimulatory receptor CD28 linked to the CD3ε endodomain (Figure 3d) [197]. These CD28ε TCRs efficiently mitigate mispairing (like CD3ζ TCRs [162]), mediate co-stimulation and preserve endogenous TCR function. Interestingly, despite reduced recognition of the cognate pMHC complex, CD28ε TCR T cells demonstrated enhanced in vivo expansion and persistence in a mouse model of melanoma. Alternatively, CD28 co-stimulation can be delivered when using the single chain TCR configuration [164].

Another strategy is to combine a transgenic TCR with a co-stimulatory CAR (coCAR), to provide signal 2 (Figure 3d). CoCARs are designed in a way similar to CARs, but lack the CD3ζ endodomain, do not kill target cells, and have previously been evaluated in combination with conventional CARs [198,199]. Full T cell activation will only occur when both the pMHC targeted with the TCR and the cell surface antigen targeted with the coCAR are encountered in close proximity. CoCAR approaches are currently being developed for safe and enhanced targeting of hematologic malignancies by providing engineered co-stimulation upon recognition of lineage antigens CD19 or CD38, which are either co-expressed on the targeted tumor cells, or expressed on cells in the malignant bone marrow microenvironment [200,201]. The CD19 coCAR enhanced T cell expansion, persistence and anti-tumor function of transgenic survivin-specific TCR T cells targeting leukemia, as well as native EBVSTs targeting lymphomas in vitro and in vivo [200]. The CD38 coCAR enhanced NY-ESO-1-specific TCR T cells, and mediated enhanced cytotoxicity against MM [201].

Lastly, a simple yet effective strategy is to increase the constitutive expression of co-stimulatory receptors in TCR T cells (Figure 3d). For instance, forced expression of 4-1BB in TCR T cells enhanced anti-tumor function in vitro and in ovo, increased cytokine production upon tumor challenge, as well as T cell expansion, proliferation and persistence [202].

#### 4.3.3. Delivery of Cytokine Signaling

Addressing the problem of the limited persistence of adoptively transferred T cells, our group developed a strategy that combines both essential co-stimulatory and cytokine signaling into one transgene by expressing the thrombopoietin receptor (c-MPL) in TCR-transgenic T cells. This approach exploits the native signaling characteristics of c-MPL (Figure 3d) in a ligand-dependent manner [203]. Thrombopoietin (TPO) is a soluble microenvironment factor in the bone marrow of patients with hematologic malignancies [232], and thus c-MPL+TCR+ T cells are stimulated through c-MPL in a microenvironment-specific manner. In addition, c-MPL+TCR+ T cells can also be pharmacologically activated by the administration of clinically available c-MPL agonists (e.g., eltrombopag). We found that c-MPL signaling in TCR+ T cells has several beneficial effects, such as enhancing T cell proliferation, expansion and cytokine production, immune synapse formation, long-term killing activity in vitro and in vivo, and preservation of a less differentiated phenotype. In addition to the known c-MPL signaling pathways, we identified by RNA sequencing that the activation of type I interferon pathways is critically involved in the c-MPL mediated immune stimulation.

Enhancing the production of cytokines by TCR-engineered T cells (Figure 3d) also serves as a promising strategy to locally modulate the immunosuppressive TME, in favor of local tumor-specific T cell persistence and anti-tumor function. For instance, overexpression of IL-12 has been explored in EBVSTs, TCR transgenic T cells and TILs [204,205,206,207,209]. These studies showed that elevated IL-12 production in TCR T cells enhanced their anti-tumor function, but also resulted in some toxicities that could be overcome if IL-12 was produced under the control of the nuclear factor of activated T cells (NFAT) or tetracycline-controlled transcriptional activation (TET)-On promoter [206,207,209]. Another study that engineered TCR transgenic T cells to overexpress IL-18 showed better persistence and tumor control when compared to IL-12 overexpression [233]. Other cytokines, including IL-15, IL-21, or constitutively active IL-7 receptor (IL-7R) signaling, have been explored in CAR T cells [210,234].

#### 4.3.4. Reverting Immune Inhibitory Signals

A well-studied strategy to mitigate immunosuppressive cytokine signaling and enhance T cell persistence in the TME is to co-express dominant negative receptors (DNRs) (Figure 3e). DNRs can be derived from truncated immunosuppressive receptors which lack an intracellular signaling domain, and capture inhibitory molecules without initiating negative regulatory signals on T cells. For example, the integration of a truncated TGF-β receptor 2 (TGFBR2) to native or transgenic TCR T cells conferred resistance to TGF-β mediated immunosuppression, and enhanced proliferation, cytokine production and anti-tumor function in models of EBV-associated lymphoma, prostate carcinoma or melanoma [211,212,213,214,215,216]. A DNR based on a truncated Fas receptor was able to mitigate pro-apoptotic FasL that is highly expressed in the TME. Expression of the Fas DNR enhanced the persistence and anti-tumor function of mouse pmel-I T cells after ACT in mice with melanoma [235].

When specifically targeting the TME, another approach is to design novel types of chimeric receptors that respond to immunosuppressive ligands expressed on cells in the TME. These receptors are called chimeric switch receptors (CSRs) and convert a T cell inhibitory into a T cell stimulatory signaling output (Figure 3e) [236]. In combination with transgenic TCRs, several CSRs have been evaluated with CTLA-4 [217], PD-1 [218,219], TIGIT [222], CD200R [220] and TGFBR2 [221,223]. All these CSRs exploit the extracellular domain of inhibitory receptors and are linked to the intracellular domain of co-stimulatory receptors, activated upon engagement of the inhibitory ligand. The CSR based on CTLA-4 (CTLA-4:CD28) endowed mouse pmel-I and OT-1/2 T cells with enhanced IFN-γ and IL-2 production, particularly in CD4+ T cells. Interestingly, in-vivo anti-tumor function of pmel-I and OT-1/2 T cells was improved by CTLA-4:CD28 expression, but was highly dependent on the presence of CD4+ T cells, and was linked to their elevated production of IL-2 [217]. The CSR based on PD-1 (PD1:CD28) enhanced the effector cytokine production of human MART-1 and p53 TCR T cells upon stimulation, and increased their proliferation and tumor control in a mouse xenograft model of human melanoma [218]. Interestingly, addition of 4-1BB as a second co-stimulatory domain (PD-1:CD28-4-1BB), reminiscent of third generation CAR T cells, did not enhance but reduced cytokine production of MART-1 TCR T cells. A similar study using a PD-1:CD28 CSR construct showed that its co-expression is equally beneficial for CD4+ and CD8+ T cells, and can improve the functional avidity of TCRs targeting tyrosinase in xenograft models of human melanoma [219]. Enhanced in vivo anti-tumor activity and cytokine production was also found in human MART-1 TCR T cells expressing a CSR based on TIGIT (TIGIT:CD28) in a human melanoma xenograft mouse model [222]. Co-expression of a CD200R-based CSR (CD200R:CD28) improved mouse and human TCR T cell function in leukemia models [220]. Finally, a TGFBR2-based CSR, which signals through 4-1BB (TGFBR2:4-1BB), enhanced the anti-tumor function of prostate cancer targeted CAR T cells [221], as well as NY-ESO-1 TCR+ T cells for melanoma [223]. Interestingly, decoy receptor variants with truncated signaling domains of most inhibitory receptors (CTLA4, PD1, TIGIT and CD200R) except for TGFBR2, did not result in significant improvement of T cell function.

Chimeric cytokine receptors (CCR) allow the redirection of an immune inhibitory cytokine input into an immune stimulatory cytokine output in engineered T cells (Figure 3e). A CCR redirecting immunosuppressive, Th2-polarizing IL-4 input into IL-7R signaling output has been developed for CAR T cells [224], and the effects in native TCR T cells has been evaluated in EBVSTs [225]. The IL-4/7 CCR rendered EBVSTs resistant to IL-4 signaling, and greatly enhanced tumor control in an EBV lymphoma xenograft mouse model, where T cells can be activated in a locally restricted manner in the TME. Future approaches will likely combine several CCRs to maximize the persistence and adaptability of engineered T cells to multiple inhibitory signal inputs in the TME. For instance, combination of a first generation prostate cancer targeting CAR (signal 1) with a TGFBR2:4-1BB CSR (signal 2) and a IL-4/7R CCR (signal 3) has produced promising results in a mouse model of pancreatic cancer, by splitting engineered T cell signals in a TME specific manner [221].

## 5. Therapeutic Genome Editing

Many T cell engineering approaches today still use viral vectors to deliver the transgenes of interest. However, the field of precision genome editing now offers the possibility of precisely integrating transgenes into specific genomic loci, while retro- or lentiviral transductions produce random integrations with variable transgene copy numbers and expression levels [237,238]. The interests of genome editing in the field of engineered T cell therapies are manifold, including (1) the generation of fully allogeneic universal donor T cell products by the elimination of endogenous TCR and HLA characteristics (e.g., universal donor CD19 CAR T cells or UCART19) [227,239], (2) the elimination of endogenous TCRs to enhance the expression and function of introduced transgenic TCRs and avoid cross-reactivities [166,168], (3) the elimination of immune inhibitory receptors to engineer resistance to immune checkpoint ligands [226,227,228,229,230,231], (4) the insertion of CAR or TCR transgenes into the TCR locus, to harness the physiological regulation of transgene expression and simultaneous elimination of endogenous TCR specificities [240,241,242], and (5) the discovery of novel targets for enhanced combinatorial immune engineering [223,243,244].

### 5.1. Elimination of Endogenous TCR Specificities

Several genome editing tools based on programmable nucleases have been developed for the targeted disruption of the endogenous TCR locus (Figure 3f) by inducing double strand breaks and DNA repair by non-homologous end joining (NHEJ). These tools include meganucleases, zinc-finger nucleases (ZFNs), transcription activator-like effector nucleases (TALENs), megaTAL nucleases, and most recently, clustered regulatory interspaced short palindromic repeat/CRISPR-associated protein 9 (CRISPR/Cas9) (reviewed in [245,246,247]). Cas nucleases are guided to the targeted DNA sequence by short guide RNAs, recognize the target DNA through Watson–Crick base pairing, and allow for multiplexing, while the other enzymes rely on protein–DNA interaction for target specificity.

ZFN-based disruption of both the two endogenous TCRβ constant (TRBC1 and TRBC2) and the TCRα constant (TRAC) regions, followed by lentiviral transduction with a transgenic WT-1-specific TCR-produced tumor, redirected T cells without endogenous TCR specificities [166]. Compared to TCR+ T cells generated by lentiviral transduction alone, the TRAC/TRBC edited TCR+ T cells were characterized by enhanced avidity, anti-tumor function and sharply reduced non-specific alloreactivity in vitro and in vivo. This seminal paper provided the proof of concept that disruption of the endogenous TRAC and TRBC loci significantly enhances the safety (by abrogating TCR mispairing), avidity and anti-tumor function of TCR locus edited T cells. Subsequently, the same group showed that ZFN-based single editing of the TRAC locus and lentiviral TCR transduction was sufficient to produce safe and highly active tumor-targeted T cells, with a protocol more amenable to clinical translation [248]. TALENs, meganucleases, megaTAL nucleases and the CRISPR/Cas9 system have also been explored for the efficient generation of TRAC and/or TRBC locus edited T cells [167,249,250]. TRAC locus disruption, comparing the various platforms, was evaluated for clinical grade manufacturing scalability, efficiency and off-target activities [250], and further optimized by others [251]. CRISPR/Cas9 mediated TRBC1/2 disruption, and lentiviral transduction with a transgenic γδ-TCR, also resulted in a more efficient anti-tumor function of γδ-TCR redirected cells, compared to lentiviral transduction alone [252].

### 5.2. Multiplexed CRISPR/Cas9-Based Genome Editing

In order to generate allogeneic universal donor TCR or CAR T cells that are resistant to PD1 mediated inhibition, a multiplexed CRISPR/Cas9-based editing approach was developed by simultaneously disrupting the TRAC, TRBC, β2-microglobulin (B2M) and PDCD1 loci (Figure 3f), combined with lentiviral transduction of a CAR [253,254]. The combination with PD1 disruption further improved the anti-tumor activity of the edited CAR T cells in preclinical models. Disruption of other immune checkpoints, including CTLA-4 alone and in combination with PD-1, as well as LAG-3, has been investigated in CAR T cells [228,229,230].

A first in human phase I clinical trial with CRISPR/Cas9-engineered NY-ESO-1 TCR-specific T cells, for the treatment of patients with advanced refractory sarcoma and MM, was recently reported [89]. The two main goals of the trial were (1) to increase the safety and efficacy of NY-ESO-1 TCR T cells, and (2) to limit the development of T cell exhaustion, compared to the previous trials with NY-ESO-1 TCRs and conventional lentiviral transduction [4,6]. For this purpose, the TRAC, TRBC and PDCD1 loci were disrupted with a CRISPR/Cas9 multiplex approach to reduce the potential for mixed TCR heterodimer formation and evade checkpoint ligand mediated T cell immunosuppression in the TME. The trial demonstrated in three patients that multiplexed genome editing of T cells with CRISPR/Cas9 and lentiviral transduction is feasible and safe. The persistence of infused T cells exceeded previous clinical results with lentivirally transduced NY-ESO-1 TCR+ T cells in the absence of editing, and potential rejection due to preexisting immune responses to Cas9 was not observed. Off-target editing has not produced safety-related side effects to date, but the patients will require a longer follow up. These promising results clearly warrant further investigation.

### 5.3. Targeted Transgene Delivery into Defined Genetic Loci

Homology directed DNA repair (HDR) can be exploited to specifically insert donor DNA templates, encoding sequences of interest, into the targeted genomic loci [247]. For example, CRISPR-based targeted integration of a CD19 CAR into the TRAC locus with adeno-associated virus 6 (AAV6) resulted in uniform CAR expression, the enhanced in vitro and in vivo anti-tumor function of CAR T cells, and simultaneous elimination of endogenous TCR specificities [240]. Placing the CAR transgene under the control of the endogenous TCR promoter lead to a more physiologic CAR expression and turnover upon antigen stimulation, as compared to constitutively active promoters, delaying the development of terminally differentiated exhausted CAR T cells [240]. MegaTAL and AAV6 mediated CAR insertion into the TRAC or the CCR5 loci have also been reported [255,256].

Insertion of tumor-targeted TCRs into the TRAC locus (Figure 3f) was realized with non-viral DNA donor templates [241,242], which significantly improves the flexibility of the approach as compared to AAV6 vectors. However, HDR success rates are lower, and require additional optimization [257]. Five different TCRs with anti-viral or anti-tumor specificities were investigated [242]. Interestingly, with this approach, TRAC locus editing alone did not completely abrogate the formation of mixed TCR dimers, and additional TRBC locus disruption was necessary. Importantly, the orthotopic insertion of a transgenic TCR produced near-physiological TCR regulation dynamics in all five different TCRs evaluated, underlining the high reproducibility of those results across different TCRs.

CRISPR tools can also be applied to engineer artificial T cell signaling circuits, to enhance anti-tumor T cell function by exploiting endogenous transcriptional regulation of specific responses (Figure 3f). For example, insertion of IL-12p70 into the IL-2Rα or PDCD1 gene locus produced T cells with antigen-dependent IL-12p70 production, resulting in enhanced anti-tumor function without significant toxicities [258]. Thus, hijacking the transcriptional regulatory system of defined loci can enable tight control of transgene expression, and could enhance the safety and specificity of functional T cell outputs.

### 5.4. Discovery of Novel Immune Engineering Targets

Engineering T cells with the capacity to persist and function long term upon adoptive transfer in vivo is one of the major goals in the field of engineered T cell therapies. Analysis of vector integration sites in CAR T cells persisting long term after infusion in patients with complete tumor responses is an elegant approach, that may reveal genetic loci of interest for targeted integration in the future. For instance, systematic analysis of vector integration sites in CD19 CAR T cells infused to patients with chronic lymphocytic leukemia (CLL) revealed that CAR T responders could be distinguished from non-responders by the pattern of genes disrupted by vector integration [259], but gene expression in pre-infusion T cells also differed between complete responders and non-responders [260]. For example, complete remission in a CLL patient with a unique genetic background (hypomorphic mutation in his other Tet methylcytosine dioxygenase 2, TET2 allele) has been ascribed to a single CD19 CAR T cell clone with a disrupted TET2 locus as a result of lentiviral insertion of the CAR transgene [261]. These T cells were characterized by an early T cell differentiation phenotype, that was previously associated with long term persistence in vivo. Another locus disruption that promoted in vivo expansion in this study was in the TGFBR2 locus. Whether differential vector integration is favored by the differential gene expression of activated T cells from responding versus non-responding patients is still an open question. Nevertheless, analysis of vector integration sites, and longitudinal analyses correlated with clinical outcomes, provide deep insights into mechanisms of T cell persistence and function, and open new avenues of investigation for enhancing engineered T cell therapies.

Tools for the broad screening of human T cells under controlled experimental conditions, that allow the identification of regulators of T cell stimulation and immune suppression, have only recently emerged [243,244]. For instance, the combination of large CRISPR-based knockout panels with in vitro killing assays revealed established (*TCEB2*, *SOCS1* and *CBLB*) and novel genetic targets (*RASA2*) for enhancing the anti-tumor function of TCR transgenic T cells [243]. Combination with an in vivo T cell infiltration screen revealed a promising target (*Dhx37*), and new functional insight into previously unrecognized immunoregulatory functions. The knockout of *Dhx37* in TCR transgenic T cells led to improved tumor control in a mouse model of triple-negative breast cancer [244]. Another example is the recently described MR1-restricted TCR, where the use of a CRISPR screen enabled the identification of MR-1 as the TCR restricting element [52].

It was also demonstrated that CRISPR-based multiplexing can be used for the generation of pooled knockin libraries, to select for the most promising novel immunostimulatory transgenes in TCR transgenic human T cells, based on functional readouts [223]. Combined in vitro and in vivo screening revealed the most promising CSR, such as TGFBR2-4-1BB, which enhanced the anti-tumor function of NY-ESO-1 TCR+ T cells in a human melanoma xenograft mouse model. Genome-wide CRISPR-based screening studies in model systems of ACT will likely also lead to the identification of novel targets in the future.

## 6. Conclusions

Basic, translational and clinical research on TCR-based ACTs has produced remarkable insight into their biology, and led to meaningful clinical responses in a variety of cancers. The field is now poised to move these therapies to the next level, as new strategies and technologies become available. The choice of a suitable target antigen and the transgenic TCR sequence are still key to success, and thus, these areas continue to be heavily investigated. Improved preclinical TCR screening is likely to enhance the safety of TCR transgenic T cell therapies, but genetic safety systems are now also well-established and can be incorporated for clinical applications. Additional T cell engineering to further enhance engineered T cells at various levels has generated intriguing results in preclinical models, including: (1) modulation of functional avidity, (2) development of MHC-independent strategies, and (3) targeting the TME (enhancing T cell homing, infiltration, proliferation, persistence, effector function and modulation of TME components). Future developments will likely harness combinatorial strategies to overcome the multitude of challenges posed by the tumors. Exploiting the tools of genome engineering will allow for even faster discovery and validation of novel approaches. The precise modification of genetic circuits will open new possibilities for controlling transgenic T cell function, and the first therapeutic genome editing applications, targeting defined genetic loci in T cells, have already reached the clinic. We are convinced that some of these novel developments have the potential to lead to clinical breakthroughs, as we learn how to best manipulate the human immune system for the fight against cancer.

## Figures and Tables

**Figure 1 cells-09-01485-f001:**
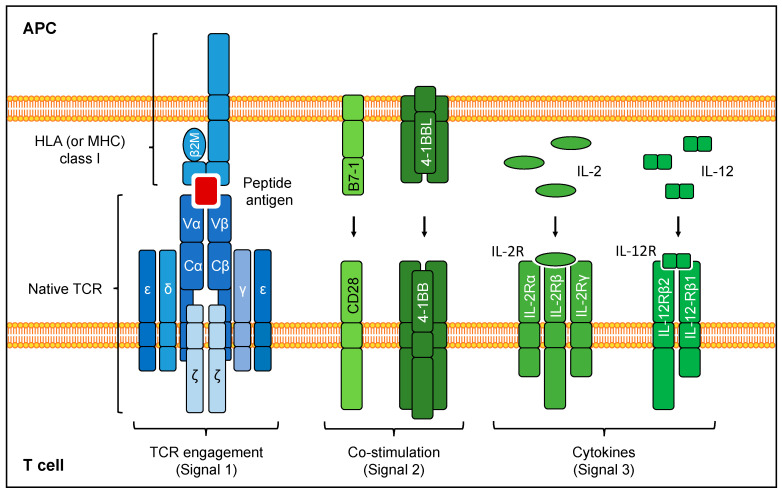
TCR:pMHC complex and signals leading to full T cell activation.

**Figure 2 cells-09-01485-f002:**
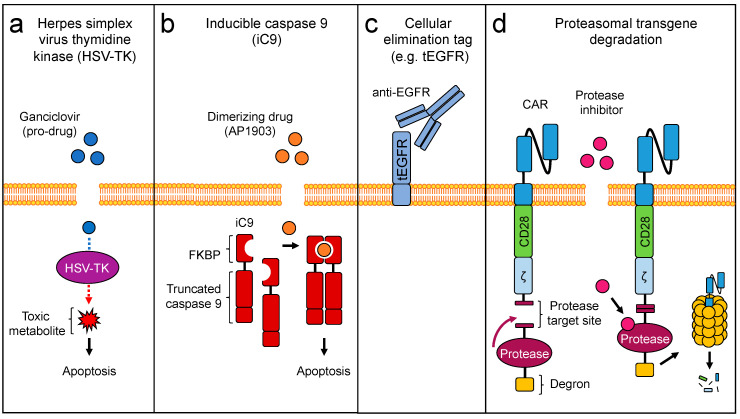
Engineering strategies to enhance safety of ACTs: (**a**) Herpes simplex virus thymidine kinase (HSV-TK). (**b**) Inducible caspase 9 (iC9). (**c**) Cellular elimination tag (e.g., tEGFR) (**d**) Proteasomal transgene degradation.

**Figure 3 cells-09-01485-f003:**
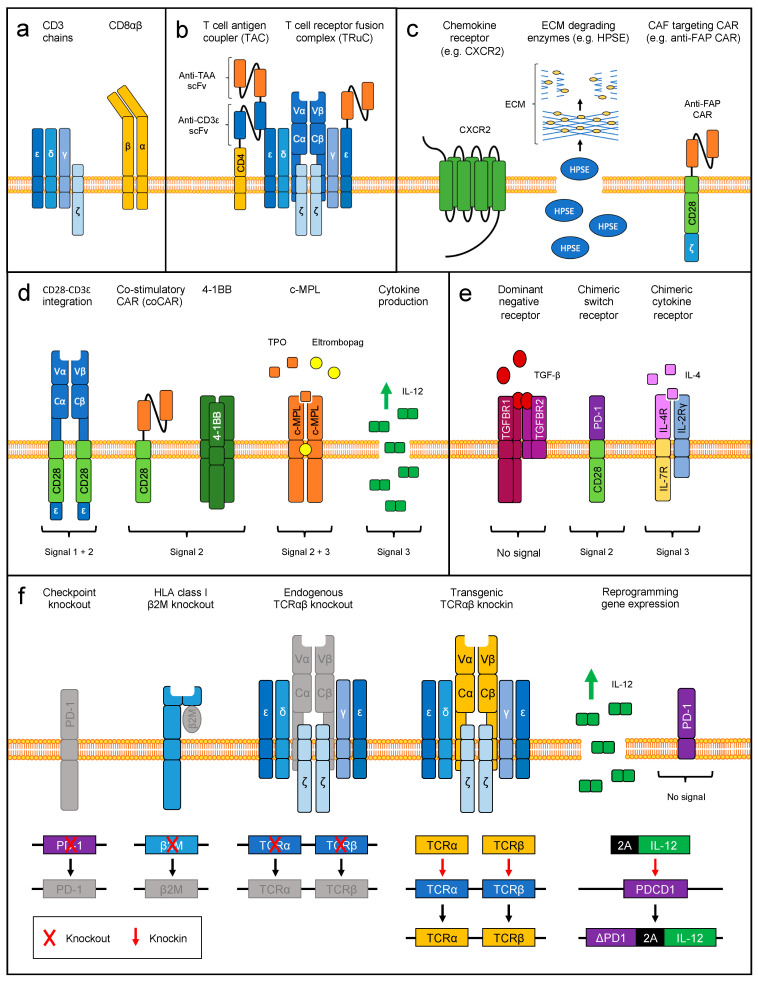
Engineering strategies to enhance function of TCR transgenic T cells: (**a**) Co-expression of TCR signaling components (CD3 and CD8αβ). (**b**) Engineering MHC-independent antigen specificity with signaling through the TCR. (**c**) Engineering improved T cell homing and infiltration into the tumor. (**d**) Engineering the delivery of co-stimulatory and cytokine signals in the TME. (**e**) Engineering the reversion of immune inhibitory signals in the TME (**f**) Genome editing strategies to improve T cell function.

**Table 1 cells-09-01485-t001:** Completed clinical trials with TCR transgenic ACT.

Antigen/ HLA	Cancer	Protocol	TCR Origin	Name	Pts	Response	Toxicities	Reference
MART1/ A*02:01	Melanoma	Lymphodepletion (Cy/Flu), ACT, high dose IL-2, peptide vaccine	Human TILs [94]	MART1	17	2 PR, 15 NR	None	[81]
MART1/ A*02:01 gp100/ A*02:01	Melanoma	Lymphodepletion (Cy/Flu), ACT, high dose IL-2	Human TILs, vaccinated mice [95]	DMF5 gp100	20 16	6 PR, 14 NR 1 CR, 2 PR, 13 NR	AE: 11× grade 2, 8× grade 3 AE: 10× grade 2, 1× grade 3 (skin, eye, ear)	[82]
MART-1/ A*02:01	Melanoma	Lymphodepletion (Cy/Flu), ACT (D0), high dose IL-2, DC vaccine (D1, D14, D30)	Human TILs [95]	DMF5	14	7 SD, 6 PD, 1 N/A	Frozen: None Fresh: SAEs: 2× grade 4 (ARDS)	[83]
NY-ESO-1/ A*02:01	Sarcoma, Melanoma	Lymphodepletion (Cy/Flu), ACT, high dose IL-2	Human TILs, affinity-enhanced [73]	NY-ESO-1 1G4-a95:LY	6 11	Sarcoma: 4 PR, 2 PD Melanoma: 2 CR, 3 PR, 6 PD	None	[4]
NY-ESO-1/ A*02:01	Sarcoma, Melanoma	Lymphodepletion (Cy/Flu), ACT, high dose IL-2, some vaccinated	Human TILs, affinity-enhanced [73]	NY-ESO-1 1G4-a95:LY	18 20	Sarcoma: 1 CR, 10 PR, 7 NR Melanoma: 4 CR, 7 PR, 9 NR	None	[5]
NY-ESO-1/ A*02:01	Multiple myeloma	High dose Melphalan + ASCT, ACT (D2), PCV vaccine (D14, 42, 90), low dose lenalidomide (start D100)	Human TILs, affinity-enhanced [73]	NY-ESO-1c259	20	14 nCR/CR, 2 VGPR, 2 PR, 1 SD, 1 PD	SAE: 7× grade 3–4, AE: 17× grade 1–3	[6]
NY-ESO-1/ A*02:01	Sarcoma	Lymphodepletion (Cy/Flu), ACT	Human TILs, affinity-enhanced [73]	NY-ESO-1c259 SPEAR T cells	12	1 CR, 5 PR, 6 SD	AE: 11× grade 3–4, CRS in 5 pts (2× grade 1, 1× grade 2, 2× grade 3)	[87]
NY-ESO-1/ A*02:01	Sarcoma	Lymphodepletion (Cy/Flu or Cy), ACT	Human TILs, affinity-enhanced [73]	NY-ESO-1c259 SPEAR T cells	42	1 CR, 14 PR, 24 SD, 3 PD	Not described	[88]
NY-ESO-1/ A*02:01	Sarcoma, Multiple Myeloma	Lymphodepletion (Cy/Flu), ACT	Vaccinated patient [96]	8F TCR NYCE T cells	3	2 SD, 1 PD	AE: 20× grade 3–4	[89]
MAGE-A3/A12/ A*02:01	Sarcoma, Melanoma, Esophageal cancer	Lymphodepletion (Cy/Flu), ACT, high dose IL-2	Vaccinated mice, affinity-enhanced [92]	MAGE-A3	9	1 CR, 4 PR, 4 NR	SAE: 2× grade 5 neurotoxicity, 1× grade 4 neurotoxicity attributed to TCR T cells	[79]
MAGE-A3/ A*01:01	Melanoma, Multiple myeloma	Lymphodepletion (Cy), Split ACT D5 (30%) D6 (70%), high dose Melphalan + ASCT, ACT (D2)	Vaccinated patient, affinity-enhanced [78]	MAGE-A3a3a	2	Not evaluable	SAE: 2× grade 5 cardiac toxicity attributed to TCR T cells	[77]
MAGE-A4/ A*24:02	Esophageal cancer	ACT (D0), peptide vaccine (D14, D28)	Human healthy donor [97]	MAGE-A4	10	7 PD, 3 SD	None	[85]
CEA/ A*02:01	Colorectal cancer	Lymphodepletion (Cy/Flu), ACT, high dose IL-2	Vaccinated mice, affinity-enhanced [91]	CEA-reactive TCR	3	2 NR, 1 PD	SAE: 2× grade 3 diarrhea, DLT, 3× inflammatory colitis attributed to TCR T cells	[84]
WT1/ A*02:01	AML	Allogeneic HCT, Prophylactic ACT if NED (D47-190), low dose IL-2, second ACT (in 7 pts)	Human healthy donor [98]	TCR-C4	12	12 CR	SAE: 2× grade 3 CRS, 24× grade 3–4 cytopenias, cGVHD: 6×, aGVHD: 2× grade 2, 1× grade 3	[9]
WT-1/ A*24:02	AML MDS	ACT (D0 and D28), peptide vaccine (D30 and D44)	Human healthy donor [99]	WT-1	8	1 SD, 3 blast reduction, 4 PD	None	[90]
MAGE-A3/ DPB1*0401	Metastatic/ refractory cancer	Lymphodepletion (Cy/Flu), ACT, high dose IL-2	Vaccinated patient [100]	MAGE-A3 (6F9 TCR)	17	1 CR, 3 PR, 13 NR	AE: 11× grade 2, 5× grade 3, 4× grade 4	[86]
HPV16 -E6/ A*02:01	Metastatic HPV16+ cancer	Lymphodepletion (Cy/Flu), ACT, high dose IL-2	Human TILs [101]	E6 TCR	12	2 PR, 4SD, 7 PD (1 pt treated 2x)	AE: 68× grade 3–4, cytopenias and IL-2 side-effects	[8]

Abbreviations. Pts: Patients; MDS: Myelodysplastic syndrome; Cy/Flu: Cyclophosphamide/Fludarabine; PCV: Pneumococcal conjugate vaccine; NED: No evaluable disease; CR: Complete response; nCR: Near complete response; PR: Partial response; VGPR: Very good partial response; NR/SD: No response/stable disease; PD: Progressive disease; AE: Adverse event; SAE: Serious adverse event; CRS: Cytokine release syndrome; ARDS: acute respiratory distress syndrome; ASCT: Autologous hematopoietic stem cell transplantation; aGVHD/cGVHD: acute/chronic graft-versus-host disease; DLT: Dose limiting toxicity.

**Table 2 cells-09-01485-t002:** Overview of engineering strategies targeting the TME. The overall goal of the listed strategies is to enhance tumor infiltration, and the persistence and function of adoptively transferred engineered T cells, by targeting and modulating TME components.

Goal of Engineering	Modification/Construct	Target of Modification	Mechanism of Action	References
Enhance tumor infiltration	Enforce chemokine receptor expression	Chemokines in the TME	Enhance infiltration into the tumor via improved detection of homing signals secreted by cells of the TME	[189]
	Enforce expression of ECM degrading enzymes	ECM	Degradation of ECM components improves T cell penetration into the tumor and migration through the tumor stroma	[193]
	CAR targeting the tumor stroma	Tumor stroma	Destruction of cellular components of the tumor stroma enables better T cell infiltration and anti-tumor function	[194,195,196]
Provide co-stimulation (signal 2)	CD28-CD3*ε* integration into TCRαβ chains	No additional recognition	Provide co-stimulation upon pMHC binding	[197]
	Co-stimulatory CAR (coCAR)	Cell surface antigen on the tumor target or bystander cell	Provide co-stimulation upon antigen binding of coCAR and pMHC recognition by TCR (both antigens required for full activation) Enhance safety and tumor specificity	[200,201]
	Co-stimulatory receptors (e.g., 4-1BB)	Cell surface co-stimulatory ligand recognition on tumor target or bystander cell	Provide co-stimulation upon binding of co-stimulatory receptor and pMHC recognition by TCR (both antigens required) Enhance safety and tumor specificity	[202]
Provide cytokine signals (signal 3)	Natural receptor targeting a microenvironment factor (e.g., c-MPL)	Soluble factor recognition in the TME (e.g., thrombopoietin, TPO)	Provide co-stimulation (signal 2) and cytokine signaling (signal 3) upon recognition of TPO and pMHC recognition by TCR (both antigens required) Act as a sink and reduce TPO levels in bone marrow microenvironment, decrease growth factor support for myeloid malignancies Remote controlled activation by c-MPL agonist drug possible Enhance safety, tumor specificity and control of action	[203]
	Enforce secretion of effector cytokines (e.g., IL-12, IL-18)	No additional recognition	Enhance engineered T cell function and persistence Modulate composition of the TME	[204,205,206,207,208,209]
	Constitutively active cytokine receptor (e.g., IL-7R)	No additional recognition	Enhance engineered T cell function and persistence	[210]
Revert immune inhibition	Dominant negative receptors (DNR)	Inhibitory signals in the TME (e.g., TGF-β)	Provide resistance to immunosuppressive factors or death ligands in the TME Reduce levels of soluble factors by acting as a sink Modulate composition of TME	[211,212,213,214,215,216]
	Chimeric switch receptors (CSR)	Inhibitory signals in the TME (e.g., checkpoint ligands)	Convert TME inhibitory immune checkpoint into co-stimulatory signals in engineered T cells upon antigen binding of CSR and pMHC recognition by TCR (both antigens required) Enhance safety and tumor specificity	[217,218,219,220,221,222,223]
	Chimeric cytokine receptors (CCR)	Inhibitory cytokines in the TME (e.g., IL-4)	Convert inhibitory soluble signals in the TME into stimulatory cytokine signals (signal 3) upon binding of CCR	[224,225]
	Knockout of checkpoint receptors	No additional recognition	Provide resistance to inhibitory immune checkpoint signals in the TME	[226,227,228,229,230,231]

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
