# Peer review of "Engineering Strategies to Enhance TCR-Based Adoptive T Cell Therapy"

_cells, 2020, doi:10.3390/cells9061485_

Round 1
Reviewer 1 Report
The manuscript entitled “Engineering strategies to enhance TCR based adoptive T cell therapy” by Rath and Arber tried to cover current status of TCR based adoptive T cell therapy, their challenges and suggest the next TCR gene therapy strategy. This review sounds interesting and educative, however there are some issues to be articulated to be published as follows:
- Authors focused on TCR based adoptive T cell therapy and its challenges and introduce some engineering strategies. Please revise abstract sound more focused and impact based on the content in the text.
- I recommend author to introduce first regarding TCR based adoptive T cell therapy. The principle and how they play anti-tumoral activity. Maybe better include historical story. Adding the Figure would be Then author may be able to lead the antigen selection and other approaches.
- As for engineering strategy, Figure 2 is useful though. I recommend author to include another figure how each engineering will modulate TME to enhance TCR gene therapy for easier understing.
- In the main text, authors better put more information. As conclusion say, basic, translational and clinical research over the last two decades’ stories and their challenges should be included main text with more detailed information with mechanism and/or principles.
- Authors can add the Table summarizing the suggested engineering strategies and how each strategies can be done and how they can enhance TCR gene therapy.
Reviewer 2 Report
This review article summarizes the current studies in the field of TCR-based adoptive T-cell therapy. Overall, this is an outstanding, well-written manuscript.
Major suggestions:
(1) Please discuss the use of “suicide gene” to improve the safety of ACT (I understand that most studies use CAR as the model system).
(2) As mentioned by the authors, “a suitable target antigen is still key to success” (Line 411). If we cannot identify more good antigens, other enhancements (Section 3 and 4) might become irrelevant. Please expand Section 2.1 and provide your own insights on this topic.
(3) The authors mentioned that the purpose of developing new engineering strategies was to address challenges in the clinical trials (Line 122-125. There’re some spelling errors here). However, the authors did not mention what the challenges were in details (Line 102?)? It’s important for us to know what we can learn from these successful or failed clinical trials. Please expand this topic.
Reviewer 3 Report
This is a well-written, timely review of the field. I think that there is room to improve the piece and make it more comprehensive with some minor tweaks/additions. In particular, there will be future scope to engineer T cells that exhibit improved engraftment, homing, proliferation, and/or persistence in vivo. Evidence for such is emerging from study of the vector-integration sites following successful CAR-T therapy and [yet to be published] in vivo whole genome CRISPR type studies in mouse models. It would improve and future-proof the piece to include these potential engineering strategies.
Specific suggestions:
1. TIL therapy constitutes TCR-based ACT, but this is not mentioned. The authors do a good job of ruling out discussion of combination therapies, cancer vaccines and CARs at the end of the first paragraph. They could include TIL therapy here too. However, I think it would improve the piece to minimally discuss TIL therapy in the introduction as it produced >20% complete remission in Stage IV melanoma prior to adoption of checkpoint blockade, so serves as a nice introduction to how effective ACT can be for end-stage solid cancers. Additional text could be framed in terms of TIL therapy having shown ACT can be effective but there being obvious room for engineering-based improvement.
2. In section 2.2 on TCR affinity, a study examining TCRs with a range of affinities for several different antigens seems noticeably absent (PubMed ID 25496365).
3. While the term “Functional avidity” (line 126) is adequately described, why use this term when its measurement has not involved any sort of measurement of “avidity”? Instead, studies tend to measure the sensitivity to a given peptide antigen so why not call it what it is, “antigen-sensitivity”? Also the sentence “structural avidity (monomeric TCR:pMHC binding strength on a cell)” on line 139 as is worrying as, by definition, avidity is not monomeric. Overall, it is not the “avidity” that is being engineered so I suggest that the authors tweak both title and text of section 3.1. The section itself correctly points out that T cells can be improved by aiding signalling without any need to alter affinity or avidity so this is only a minor adjustment.
4. Please remove reference 106. The idea that the CD8 coreceptor stabilises monomeric TCR-peptide-MHC interactions as reported in this Nature paper has been debunked (PMID 18243322). The results in that study are now well known to have been an SPR artefact (the response unit traces never returned to baseline indicating aggregated protein). CD8 does stabilise the interaction between the TCR and CD8 as measured using peptide-HLA multimers (genuine avidity; PMID 15837791) by altering TCR off rate.
5. Section 3 on strategies to enhance anti-tumour activity of TCR-transgenic T-cells lacks and mention of T-cell trafficking/homing. There’s also no mention of strategies to increase T-cell engraftment/persistence. Several such strategies are in the pipeline; for example, UPenn are using TET2 knockout they described (PMID 29849141). Indeed, the vector-integration site can have large effects on T-cell therapy and suggests means by which T-cells might be engineered to improve proliferation, engraftment or persistence (e.g. see PMID 31845905). “These findings provide multiple potential approaches to optimizing T cell engineering for optimal function in immunotherapy protocols” (the last line of PMID 31845905) so I think these other strategies deserve a mention here. L-selectin (CD62L) has been used to produce prolonged in vivo persistence and superior anti-lymphoma activity of NK cells (PMID 27183388). Engineering of CD62L into CD8 T-cells improved their efficiency in mouse models (PMID 31249570) and I believe that there are there plans to engineer in human CD62L for trials at the Perelman. This is not my field, so there may be other examples out there that could be included too.
6. CRISPR-mediated TCR replacement seems to represent the ultimate way in which to circumvent TCR mispairing yet it is missing from Figure 1a (instead endogenous TCR chain KO occurs in panel e). I also suggest that the PD1 knockout in Fig 1e is generalised to “checkpoint knockout” or similar with PD1 as an example. Various groups have removed other checkpoint molecules as discussed in the text, so a broader net would catch all these fish.
7. Section 3.2 on “MHC independence” should state the obvious disadvantage of MHC-restriction is that any [MHC-restricted] TCR can only ever be used to treat a minority of patients. This might be obvious to the expert, but I think it would be worth stating this major caveat of MHC-restriction in the first sentence of this section. This section also omits the fact that some TCRs recognise cancer cells without the need for specific MHC class I (e.g. PMIDs 22885985 and 31959982). I think it would be worth mentioning HLA-agnostic TCRs in the introduction to this section for the sake of completeness.
FYI: Please also be aware that the non-viral orthotopic TCR replacement as described in reference 183 is ~0.2% efficient for both alpha and beta locus in the same cell, so I do not see this approach being used in the clinic in its current form. The UPenn AAV6 approach is far more efficient (>60%).
Round 2
Reviewer 1 Report
Authors revised accordingly.